# Development of Biomaterials Based on Biomimetic Trace Elements Co-Doped Hydroxyapatite: Physical, In Vitro Osteoblast-like Cell Growth and In Vivo Cytotoxicity in Zebrafish Studies

**DOI:** 10.3390/nano13020255

**Published:** 2023-01-06

**Authors:** Tanatsaparn Tithito, Siwapech Sillapaprayoon, Wittaya Pimtong, Jirawan Thongbunchoo, Narattaphol Charoenphandhu, Nateetip Krishnamra, Aurachat Lert-itthiporn, Weerakanya Maneeprakorn, Weeraphat Pon-On

**Affiliations:** 1Department of Physics, Faculty of Science, Kasetsart University, Bangkok 10900, Thailand; 2Nano Environmental and Health Safety Research Team, National Nanotechnology Center (NANOTEC), National Science and Technology Development Agency (NSTDA), Khlong Luang 12120, Thailand; 3Center of Calcium and Bone Research (COCAB), Faculty of Science, Mahidol University, Bangkok 10400, Thailand; 4Department of Physiology, Faculty of Science, Mahidol University, Bangkok 10400, Thailand; 5Institute of Molecular Biosciences, Mahidol University, Salaya 73170, Thailand; 6The Academy of Science, The Royal Society of Thailand, Dusit, Bangkok 10300, Thailand; 7Responsive Nanomaterials Research Team, National Nanotechnology Center (NANOTEC), National Science and Technology Development Agency (NSTDA), Khlong Luang 12120, Thailand

**Keywords:** hydroxyapatite, biomimetic trace elements, ions-doped HA, bone tissue engineering, zebrafish

## Abstract

Synthesized hydroxyapatite (sHA)—calcium phosphate (CaP) based biomaterials play a vital role and have been widely used in the process of bone regeneration for bone defect repair, due to their similarities to the inorganic components of human bones. However, for bone tissue engineering purpose, the composite components, physical and biological properties, efficacy and safety of sHA still need further improvements. In this work, we synthesized inhomogeneous hydroxyapatite based on biomimetic trace elements (Mg, Fe, Zn, Mn, Cu, Ni, Mo, Sr, Co, BO_3_^3−^, and CO_3_^2−^) co-doped into HA (THA) (Ca_10−*δ*_M*_δ_*(PO_4_)_5.5_(CO_3_)_0.5_(OH)_2_, M = trace elements) via co-precipitation from an ionic solution. The physical properties, their bioactivities using in vitro osteoblast cells, and in vivo cytotoxicity using zebrafish were studied. By introducing biomimetic trace elements, the as-prepared THA samples showed nanorod (needle-like) structures, having a positively charged surface (6.49 meV), and showing paramagnetic behavior. The bioactivity studies demonstrated that the THA substrate can induce apatite particles to cover its surface and be in contact with surrounding simulated body fluid (SBF). In vitro biological assays revealed that the osteoblast-like UMR-106 cells were well-attached with growth and proliferation on the substrate’s surface. Upon differentiation, enhanced ALP (alkaline phosphatase) activity was observed for bone cells on the surface of the THA compared with that on the control substrates (sHA). The in vivo performance in embryonic zebrafish studies showed that the synthesized THA particles are nontoxic based on the measurements of essential parameters such as survivability, hatching rate, and the morphology of the embryo. The mechanism of the ions release profile using digital conductivity measurement revealed that sustained controlled release was successfully achieved. These preliminary results indicated that the synthesized THA could be a promising material for potential practical applications in bone tissue engineering.

## 1. Introduction

Bioceramic materials based on the calcium phosphate of hydroxyapatite (HA) have been widely synthesized and investigated for bone tissue engineering because of their excellent biocompatibility and bioactivity including their osteoinductivity/conductivity [1,2,3,4,5]. In comparison to natural hydroxyapatite, synthetic HA (Ca_10_(PO_4_)_6_(OH)_2_) has several limitations due to differences in its chemical and in vivo implantation properties compared to those of natural bone. Generally, natural hydroxyapatite is not a homogenous material. Besides calcium (Ca^2+^) and phosphate (PO_4_^3−^) ions, natural bone incorporates various inorganic mineral elements such as Mg^2+^, K^2+^, Fe^2+^, Zn^2+^, Mn^2+^, Cu^2+^, Ni^2+^, Mo^3+^, Sr^2+^, BO_3_^3−^ and CO_3_^2−^ which act as the biomimetic ions needed for the formation of the building blocks in soft and hard tissues [6,7,8]. These ions are very important since they are found in different protein complexes (enzymes) which control physiological functions and environments and modulate the equilibrium conditions in the osteoblasts, osteoclasts, and osteocytes. In addition, these ions would directly influence the process of the bone healing process by the release effects of the biologically active ions on cellular activities to direct cellular ingrowth and consequently tissue regeneration [9,10,11,12,13]. To study the effect of biomimetic trace ions on bone tissue regeneration, Mansour, A. et al. [14], produced a CaP cement with biomimetic trace elements (such as Na^+^, K^+^, Mg^2+^, Sr^2+^, Zn^2+^, Ni^2+^, Ti^+^, Mo^+^, Cr^3+^, etc.) from natural hydroxyapatite through the calcination of bovine bone. They found that this CaP cement was superior in stimulating bone regeneration compared with the cements made from synthetic HA. However, this study did not address the important aspect of the ions release kinetics from the tested cement. Additionally, interest in trace elements to improve the physical and biological properties of HA has been reviewed by Uskoković V. et al. [6], Brokesh, A.M. et al. [7], and Glenske, K. et al. [8]. These reviews have reported that the presence of trace elements in the HA structure can cause the modulations of the immune microenvironment that direct tissue healing and regeneration. Although many of the synthetic ions co-doped with hydroxyapatite substitutes exhibit excellent results, most of the studies have only considered a few of the trace elements. The challenge of doping hydroxyapatite with synthetic multi-elements was studied by Vukomanovic, M. et al. [15]. In this work, the multi-doped apatites containing Mg^2+^, Sr^2+^, Zn^2+^, and Ga^3+^ ions (HApMgSrZnGa) were synthesized and the promoted osteogenic differentiation of MSCs was observed. Notably, better biological processes for bone regeneration mainly result from a combination of intrinsic (mechanical properties, surface charge, surface roughness, etc.) and extrinsic (biological active molecules, etc.) features of the synthetic hydroxyapatite with added trace elements. Besides using biomimetic trace elements as dopants in HA, previous investigations also revealed that HA doped with rare earth (RE) elements are also possible. The results showed that this can promote bone regeneration via stimulated stem cell adhesion [16]. On the other hand, based on the idea of improving bone-substituted materials, the preparation of Ca-deficient hydroxyapatite (CDH) decorated with oxide ceramic (such as alumina or zirconia ceramics) was developed [17,18].

To develop the synthetic ions co-doping hydroxyapatite, we synthesized inhomogeneous hydroxyapatite based on the multi-trace elements of Mg, K, Fe, Zn, Mn, Cu, Ni, Mo, Sr, BO_3_^3−^ and CO_3_^2−^ co-doped into the HA (THA) (Ca_10−*δ*_M*_δ_*(PO_4_)_5.5_(CO_3_)_0.5_(OH)_2_, M = trace elements) via co-precipitation from an ionic solution. A potential as biomaterials for bone tissue purposes, the in vitro biological assay and in vivo investigations of the effects of adding these trace elements were studied. To test the activity, the attachment, growth, and proliferation of osteoblast UMR-106 cells on the surface of THA were investigated and compared with the results obtained from control substrates, i.e., glass and sintered sHA samples. In addition, zebrafish were used in the in vivo test to assess the nanosafety of the THA [19,20]. This is because the high fecundity of the females offers the high animal numbers needed for screening experiments and the rapid development of the embryos shortens the experimental time compared to the use of rodents. In addition, the transparent body of the embryos allows optical analyses to easily see the inner organs of the tested animals. Although zebrafish have similar genetics and physiology to mammals and humans, they differ from animals (humans) in that they lack lungs, prostate, and breast tissue [21,22]. Thus, the in vivo study aiming to assess the safety of sHA and THA particles must include an additional assessment of embryonic development. This is the purpose of the additional testing involving the osteoblast-like UMR-106 cells. The osteoblast-like UMR-106 cells cultured on sHA and THA substrates show that not only the synthesized materials are biocompatible but also the osteoblast-like UMR-106 cells are well-attached and the growth and proliferation on the material surfaces are improved. It was seen that after exposing zebrafish embryos to the nanoparticles, the survival rates, hatching rates, morphological phenotypes, and heart rates were improved and no toxic behaviors were observed. In addition, we also demonstrated that the total release of ions from multiple trace elements co-doping HA was much higher than that of sHA (without ions doping).

## 2. Materials and Methods

### 2.1. Chemical Reagents

The chemicals used in this study were Ca(NO_3_)_2_·4H_2_O (calcium nitrate, 99%) (Kemaus, Cherrybrook, Australia), (NH_4_)_2_HPO_4_ (diammonium hydrogen phosphate, 99%) (Kemaus, Cherrybrook, Australia), Na_2_CO_3_ (sodium carbonate, 99.8%) (Univar, Auckland, New Zealand), MgCl_2_·6H_2_O (magnesium chloride, 99%) (Kemaus, Cherrybrook, Australia), H_3_BO_3_ (boric acid, 99.5%) (Merck, Darmstadt, Germany), FeCl_2_·4H_2_O (ferrous chloride, 99%) (Sigma-Aldrich, St. Louis, MO, USA), ZnCl_2_ (zinc chloride, 98%) (QRëC, Auckland, New Zealand), CuCl_2_·2H_2_O (copper (II) chloride dihydrate, 99%) (QRëC, Auckland, New Zealand), Mn(NO_3_)_2_·4H_2_O (manganese (II) nitrate tetrahydrate, 99.8%) (Chem-supply, Gillman, SA, Australia), NiCl_2_·6H_2_O (nickel(II) chloride hexahydrate, 99%) (Merck, Darmstadt, Germany), Na_2_MoO_4_·2H_2_O (sodium molybdate dihydrate, 99.5%) (Kemaus, Cherrybrook, Australia), SrCl_2_·6H_2_O (strontium chloride hexahydrate, 99%) (Univar, Auckland, New Zealand) CoCl_2_·6H_2_O (cobalt chloride hexahydrate, 98%) (Sigma-Aldrich, St. Louis, MO, USA), and NH_4_OH (ammonium solution, 28%) (QRëC, Auckland, New Zealand).

### 2.2. Synthesis of Multi-Trace Elements Co-Doped Hydroxyapatite (THA)

In this research, the co-precipitation method from solution was used to synthesize the multi-trace elements co-doped hydroxyapatite (THA) with the chemical formula of Ca_10−*δ*_M*_δ_*(PO_4_)_5.5_(CO_3_)_0.5_(OH)_2_, where M = trace elements. The flowchart of the preparation process is shown in Figure 1. To fabricate the multi-ions-doped hydroxyapatite, separate cations and anions solutions were prepared. The cations solution was prepared using Ca(NO_3_)_2_·4H_2_O, MgCl_2_·6H_2_O, FeCl_2_·4H_2_O, ZnCl_2_, Mn(NO_3_)_2_·4H_2_O, CuCl_2_·2H_2_O, NiCl_2_·6H_2_O, Na_2_MoO_4_·2H_2_O, CoCl_2_·6H_2_O, and SrCl_2_·6H_2_O, being the source of Ca, Mg, Fe, Zn, Mn, Cu, Ni, Mo, Co, and Sr, respectively. These chemicals were mixed in 250 mL of deionized water in a beaker under the action of a magnetic stirrer. The anionic solution was obtained by dissolving (NH_4_)_2_HPO_4_, H_3_BO, and NaCO_3_ in 250 mL of deionized water containing appropriate amounts of P with BO_3_^3−^ and CO_3_^2−^, respectively. The amounts of the elements in both solutions are shown in Table 1.

The mixing of the two solutions was performed in a beaker using a magnetic stirrer. Next, the cations and anions solutions were mixed by slowly dropping the cation solution into the anion solution. The pH of this last mixture was maintained at 9 by dropping (NH_4_)OH into the mixed solution. This was all carried out under constant stirring. After several minutes, formations of ions-doped hydroxyapatite began to co-precipitate. These precipitates then descended to the bottom of the beaker. The powders were then removed from the supernatant solutions. The resulting powders were filtered and washed with de-ionized water until the pH was about 7 and collected by freeze-drying. Before the physicochemical characterization, in vitro biological assays, and in vivo study in zebrafish, sample powder was heated to 500 °C for 2 h in an ambient air atmosphere. The ions-doped hydroxyapatite was labeled as THA. Pure hydroxyapatite (without ions) was synthesized using the same method and labeled as sHA.

### 2.3. Characterization of the Composite Scaffolds

#### 2.3.1. Physicochemical Characterizations

The morphology and chemical elements of the as-prepared sHA and THA particles were characterized using a scanning electron microscope (SEM) and energy dispersion spectroscopy (EDS) (FEI SEM Quanta 450, Brno, Czech), operating at an acceleration voltage in the range of 10–15 kV. These were used to observe the size, morphology, and elements in the samples. To determine the presence of the different phases of the synthesized particles, an X-ray diffractometer (XRD) (D8 Advance Bruker, Karlsruhe, Germany) was used to determine the crystal structure of the sHA and THA powders using CuKα radiation (λ = 0.15405 nm). The XRD spectrum was scanned over the 2θ range between 20 and 60°. The specific surface area of the sHA and THA particles was measured using the Brunauer–Emmett–Teller (BET) method at the relative pressure of P/P_0_ = 0.996, which was performed on a Micrometrics Gemini V2 model 2380 (Micromeritics Instruments Corporation, Norcross, GA, USA). Meanwhile, the surface charge of the particles was measured using a zeta potential analyzer (Zetasizer 3000HS, Malvern, UK). The magnetic properties of the sHA and THA particles were measured at room temperature using a VSM (vibrating sample magnetometer) (Lakeshore, Model 4500, Carson, CA, USA). The shape and size of the as-prepared sHA and THA at the nanoscale were examined with a transmission electron microscope (TEM) (TEM-Hitachi HI7700, Tokyo, Japan). TEM Images were taken of a one-drop dispersed solution of sHA and THA deposited on carbon grids.

#### 2.3.2. Bioactivity of the THA in SBF

The bioactivity of synthesized THA was studied in a simulated body fluid (SBF) solution which was prepared using the Kokubo method [23] in an incubator for 7 days at 37 °C. SBF is a solution with an ion concentration close to that of human blood plasma. It is made by buffering the following chemicals: NaCl (136.8 mM), NaHCO_3_ (4.2 mM), KCl (3.0 mM), K_2_HPO_4_ (1.0 mM), MgCl_2_∙6 H_2_O (1.5 mM), CaCl_2_ (2.5 mM), and Na_2_SO_4_ (0.5 mM) at a pH of 7.4 in (CH_2_OH)_3_CNH_3_ and HCl. The bioactivity of the mineralized apatite formation of each THA sample was assessed by immersing the pellets in 50 mL of SBF for one week at 37 °C. The SBF was replaced every three days to avoid any effects caused by changes in the cationic concentration that might occur due to the degradation of the sample. After soaking in SBF, the composite scaffolds were rinsed with deionized water and dried. SEM and EDS observations were used to study the mineralized apatite and chemical elements on the surface of the samples.

#### 2.3.3. Electrical Conductivity of the Ions Release Profile from THA

The release of trace elements from THA particles was studied in static conditions. Typically, 0.15 g of as-prepared THA powder was dispersed in 15 mL distilled water at a pH of 7.4 at room temperature, and at various time points, 0.5, 1, 3, 5, 7, and 14 days. At the end of aging periods, the electrical conductivity values which are related to the number of ions released at given periods were measured using a digital conductivity meter (HI9813-5 portable pH/EC/TDS meter, HANNA instruments, Woonsocket, RI, USA). To study the ion release profile between THA and sHA, conductivity measurements were plotted as a function of time.

#### 2.3.4. In Vitro Studies

The substrates for cell culture were circular glass discs (control), sHA, and THA discs with a diameter of ~10 mm and a thickness of 3 mm, all of which were sterilized with 70% ethanol and placed under a UV lamp for 30 min before use. The rat osteoblast-like UMR-106 cells (American Type Culture Collection (ATCC) No. CRL-1661) were grown in a Dulbecco’s modified Eagle’s medium (DMEM, Sigma, USA) supplemented with 10% *v*/*v* fetal bovine serum (FBS, Gibco, Auckland, New Zealand) and 100 U/mL penicillin–streptomycin (Gibco, USA) and seeded on the circular glass, sHA and THA discs at 1 × 10^6^ cells/well in 6-well plates. The well plates were incubated at 37 °C in a humidified 5% CO_2_. The medium was changed every 3 days. The examinations of the cell attachment and mineralization were performed on days 3, 5, and 7 after seeding. The glass and substrates were first rinsed with phosphate buffer (pH 7.2) twice and then fixed in 2.5% glutaraldehyde (Unilab, New South Wales, Australia) in 0.1 M phosphate buffer for 3 h. Then, the glass and substrate discs were dehydrated in a series of ethanol/water solutions (50%, 70%, 80%, and 90% *v*/*v*) and absolute ethanol. The glass and substrate discs were then desiccated under a vacuum. The cell morphology, cell adhesion, and cell proliferation were visualized by SEM.

Cell viabilities were determined using an MTT (3-(4,5-dimethylthiazol-2-yl)-2,5-diphenyl-tetrazolium bromide) assay which was carried out to determine the cytotoxicity and metabolic activity of cells on the glass, sHA and synthesized THA discs. The method has been described in our previous studies [24]. Briefly, the cells on the substrates were first incubated on the substrates being studied for 3, 5, and 7 days. The substrate scaffolds were washed and had 20 μL (0.5 mg/mL) MTT solution added to each sample and incubated for 3.5 h. Upon removal of the MTT solution, the scaffolds were smashed and the purple formazan crystals were dissolved in 150 μL of the solvent containing 4 mM HCl and 0.1% Nondet P-40 (NP40) in isopropanol while shaking the plate for 15 min. Then, the solution was centrifuged and the supernatant was transferred into another plate to measure the optical density using a microplate reader (Bio-Tek EL×800, Goleta, CA, USA) at 595 nm.

#### 2.3.5. Alkaline Phosphatase (ALP) Activity of UMR-106 Cells on sHA and THA Materials

The osteogenic differentiation was assessed by measuring the time course of the alkaline phosphatase (ALP) activity of UMR-106 cells grown on the circular glass discs (control), sHA, and THA discs. The osteogenic induction medium for UMR-106 cells was a Dulbecco’s modification of Eagle’s medium (DMEM; catalog no. D6429; Sigma, St. Louis, MO, USA) consisting of 4.5 g/L glucose, 0.584 g/L L-glutamine, 3.7 g/L NaHCO_3_, 0.11 g/L sodium pyruvate, inorganic salts (e.g., 0.2 g/L CaCl_2_ and 0.109 g/L NaH_2_PO_4_), and vitamins (e.g., 0.004 g/L folic acid, and 0.004 g/L pyridoxine). For the ALP activity determination, the substrates were first rinsed with phosphate buffer (pH 7.2). Then, 1% Triton X-100 and 1 mM MgCl_2_ were added to each well to lyse the cells. After lysing, the aliquot of the lysates was sonicated and centrifuged at 16,000 rpm for 10 min at 37 °C and incubated in 1 mg/mL *p*-nitrophenylphosphate (Sigma-Aldrich, St. Louis, MO, USA). The conversion of *p*-nitrophenylphosphate into p-nitrophenol in the presence of ALP and its absorbance were determined at 405 nm (Tecan, SpectraFluor plus, Kenilworth, NJ, USA). This process was repeated three times for each sample type and time point. 

#### 2.3.6. Zebrafish Husbandry

Adult zebrafish (Danio rerio) were raised in a recirculation system (AAB-074, Yakos65, New Taipei City, Taiwan) under a photoperiod of 14/10 h (day/night) under optimum water conditions, i.e., temperature (28.5 ± 1 °C), pH (6.0–8.0), conductivity (300–700 µS) and DO (>6 mg/L). Embryos were obtained from the mating of the fishes and were raised in egg water (0.03% (*w*/*v*) sea salt in DI water). Fertilized (normal) embryos obtained at 4 h post-fertilization (hpf) were selected for the study using a stereomicroscope (SZX16, Olympus, Tokyo, Japan). The experimental procedures used were approved by the NSTDA Institutional Animal Care and Use Committee (No. 005-2562).

#### 2.3.7. Zebrafish Embryo Acute Toxicity Test

The used zebrafish embryo acute toxicity assay follows the Organization for Economic Co-operation and Development guideline 236 (OECD 236, 2013). The sHA and THA particles were prepared as a stock solution by adding the particles to the DI water and sonicating the mixture for 10 min. For the test, the stock solutions were diluted in egg water to final concentrations of 20, 40, 60, 80, and 100 µg/mL and sonicated for 10 min. Twenty good-quality embryos were exposed with 2 mL for each test solution in a 12-well culture plate and 2 mL of egg water was used as a negative control. Each test solution was vortexed for 20 s before adding to the wells. The experiment was performed as three replicates. The plates were incubated at 28.5 ± 1 °C under a photoperiod of 14/10 h day/night for up to 96 h. Every 24 h, the test solutions were refreshed and dead embryos were removed. The number of dead, hatching, abnormal embryo and morphology were recorded at exposures of 24, 48, 72, and 96 h. The heart rates of the embryos were observed at 24 and 48 hpf by the video recording of five randomly selected embryos per replicate.

#### 2.3.8. Statistical Analysis

All data were examined using the mean ± standard deviation (SD) test from three independent experiments. The statistical significance between each test solution and control was determined by using a one-way analysis of variance (ANOVA) followed by Tukey’s test to compare the differences between groups. A *p*-value < 0.05 was regarded as being statistically significant.

## 3. Results and Discussion

The XRD diffraction patterns of the as-prepared and the sHA and THA particles calcined at 500 °C for 2 h (Figure 2a,b) exhibited the characteristic peaks of apatite (PDF. 00-064-0738), i.e., peaks at 2θ = 25° (200), 32° (211), 34° (202), and 47° (222). Additionally, the diffraction peak for THA (Figure 2b) was slightly increased by calcination. Compared with the XRD patterns of sHA (Figure 2a), the THA pattern exhibits a lower peak intensity and a broadening of the diffraction peaks. This implies the low crystallinity of THA (due to the formation of smaller crystals) (more amorphous). The analysis of the different THA patterns indicated that the crystallinity gradually decreased as more trace elements were loaded into the HA. In addition, the XRD results showed that the incorporation of the ions did not lead to the formation of the secondary phases of other calcium phosphates, such as tri-calcium phosphate (TCP) or biphasic calcium phosphate (BCP), or metal oxide phases arising from the co-doping with the trace elements.

The magnetic property of this biomaterial is another key factor that affects bone tissue regeneration [25]. The intrinsic magnetic properties of sHA and THA were investigated using a VSM at room temperature, as shown in Figure 3. From the measurement, the room temperature magnetization curve (magnetization vs. the applied field (M-H)) of the as-prepared THA was approximately linear. Thus, this sample exhibited paramagnetic-like behavior with the magnetization of 0.26 emu/g compared with that of sHA which exhibited diamagnetic behavior which was also clearly observed in previous reports on the magnetic properties of undoped hydroxyapatite [26]. This trend could be explained based on the synergistic effect of two substitutional magnetic ions (i.e., Fe^3+^, Co^2+^, and Ni^2+^). The paramagnetic nature is necessary for their desired utilization in biomedical applications, such as a contrast agent in MRI applications [25]. In addition, other reports showed that the magnetic nature of biomaterials leads to an enhancement of excellent bone cell ingrowth due to the presence of internal magnetic fields (physical cue) of materials [23,24]. Based on these reports and the result obtained in this study, it could be concluded that the multi-trace element co-doped HA (THA) has a synergistic effect on the cellular attachment process and can stimulate osteoblast activity.

The surface properties of THA compared with those of sHA, such as the surface charge and surface area, can be characterized by zeta potential measurements and the BET method. The results revealed that the positive value of zeta potential appears when the trace elements are doped into the HA (6.49 mV), while sHA produces a negative zeta potential value of −4.02 mV. The incorporation of ions into HA can lead to a change in microstructure which in turn alternate the surface charge. The positive zeta potential of THA might be due to the aggregation of the cationic nature of multi-trace elements on the substrate surfaces. The positive zeta potential of the as-prepared THA leads to an increase in the solubility of the particles and may lead to aggregation. However, it also promotes the adsorption of negatively charged biomolecules on the surface and improves the efficacy of drug delivery [26]. As previously shown, the positive and negative values of the zeta potential of biomaterials aid bone cell preferential adhesion and support cell growth, however, their morphologies are significantly different. This effect results from the fact that the cell membrane would possess an overall negative charge which would interact with a positive charge. The electrostatic repulsion between the negatively charged particles would cause protein localization resulting in a significant biological response [2,6]. Figure 4 shows the BET-specific surface area of THA and sHA. THA shows higher specific surface area (105.02 m^2^ g^−1^) than that of sHA (66.98 m^2^ g^−1^) corresponding to a pore size of 29.72 and 19.38 nm, respectively, confirming a mesoporous (5–50 nm) structure of THA and sHA. The slightly positively charged surface and higher surface area of the as-prepared THA particles can be used for protein and drug delivery applications. In addition, it might be considered a key factor for ions dissolution when in contact with the surrounding medium.

The morphology of the as-prepared THA showed that THA is formed by the agglomeration of nanorods and elongated shapes, needle-like crystals (evidenced in the SEM image seen in Figure 5c). This was confirmed by the TEM image (figure inset). This morphology is similar to those reported in previous reports [6,7,8], which also show that the ions-doped HA particles are formed as rods and have a needle-like morphology. Meanwhile, the sHA particle images (Figure 5a) exhibited shorter and wider rod shapes. They also showed that sHA has lower agglomerations of the smaller rods. Based on these phenomena, it can be implied that the ions doping restricted the formation of apatite crystal during the synthesis process. The EDS analysis (See Figure 5b,d) indicated the ion species to be Ca, P, Mg, K, B, Fe, Zn, Mn, Cu, Ni, Mo, Sr, and Co. This confirmed the presence of these elements in the synthetic particle. The lower peak intensity of trace elements in the synthesized THA particle implied that the elements were simultaneously incorporated into the HA structure during the doping process.

For the bioactivity assessment of the THA in SBF, the formation of apatite particles on the THA surface after being soaked in SBF for 7 days was examined by SEM and EDS analysis (Figure 6). The SEM image of the THA surface in Figure 6a shows the free precipitate before incubation in SBF. After immersing the scaffolds in SBF for 7 days, inhomogeneous growth of apatite particles was observed on the THA surface (Figure 6b,c). The Ca to P ratio of the particles formed on the surface, as determined by the EDS analysis, was 1.59 (see Figure 6d). This feature is attributed to the dissolution of cations such as Ca^2+^, Mg^2+^, and Sr^2+^, and PO_4_^3−^ anions acting as deposition sites of apatite to precipitate crystals in the SBF solution. The formation of mineral apatite leads to the surrounding bone being able to come into direct contact with this layer.

Since HA with or without the trace metal elements doping were exposed to the solution, degradation simultaneously releases both cations and anions. To study the ion release profile, the electrical conductivity as a function of time up to 14 days was measured. As can be seen in Figure 7, the electrical conductivity of THA in water gradually increased over time and a faster release was observed in the first seven hours. Meanwhile, the conductivity value of sHA was quite stable. During entire aging period, the electrical conductivity of THA was in the range of 620–1100 μS cm^−1^, which was higher than that of sHA (110–410 μS cm^−1^). The high initial electrical conductivity value might be attributed to the dissolution of THA continuously releasing ionic composition into the liquid solution. Generally, bulk hydroxyapatite is insensitive when exposed to water at neutral and basic pH values, and it has been reported that HA with trace metal elements doping is also chemically stable in these conditions. For the longer period of aging, the conductivity value of the solution gradually increased over time but no saturation value was attained. The dissolution of biomaterial, especially calcium phosphate-based bioceramic materials in a liquid solution, is a complex process that involves chemical reactions with the formation of different phases. Thus, no equilibrium state can be reached [27].

The results from the investigations into the non-toxicity and biocompatibility of THA, cell attachment, and proliferation on the THA surface were compared with those obtained from sHA and glass (as control substrates) after 3, 5, and 7 days of incubation. The results of the THA experiments are shown in Figure 8. The SEM images revealed that the attachment of UMR-106 osteoblast-like cells to all substrate surfaces increased over time. The cells covered and spread on glass and THA surfaces were higher than that on sHA at each time point. There was not much difference between the UMR-106 osteoblast-like cell density in the case of depositions onto the glass and THA after the incubation times were increased to 7 days. Meanwhile, in the case of sHA, the cell coverage on the surface of the sample was not perfect after 7 days of incubation. Although the cell growths on glass and THA are not significant, the cells on THA exhibit pseudo-pod-like cellular extensions (arrow) and are connected to other cells indicating the successful proliferation of cells. Previous reports have suggested that the stimulated cell growth and proliferation on the ion-doped calcium phosphate resulted from the dissolution of ions [2,9,10]. These released ions affect important components of the bone microenvironment. Based on the influence of the release of ions on cell growth, the observations revealed that the multi-ions co-doped calcium phosphate-based biomaterials better stimulated cellular viability than undoped or slightly doped groups [9,10]. In addition, the physical properties of materials such as the negative or positive zeta potential at physiological pH, high surface area, and satisfactory pore size distribution of the materials were the key factors in improving cellular attachment and proliferation [2].

The cytotoxic effect of as-prepared THA material was determined from the MTT assay and in vivo zebrafish assay. These two assays were performed, and the results are shown in Figure 9. The MTT was performed using different incubation times (3, 5, and 7 days). The results for the glass, sHA, and THA revealed an increase in cell viability as the incubation period was increased. In addition, the cell viability on the glass and THA substrates exhibited no significant difference on days 5 and 7 of the cell culture. The improved cell attachment and cell viability might be due to the fact that the biomaterials contain various bioactive ions such as Zn, Fe, Mg, and Sr which are present in the bone extracellular matrix (EMC). Moreover, these ions have been reported to modulate the osteoblasts’ activity [6,7,8,9,10]. Since the ALP activity function is an early marker for osteogenic differentiation, the ALP activity of UMR-106 cells cultured on the sHA and THA surfaces are shown in Figure 9. Similar to cell proliferation results, the alkaline phosphatase enzyme increases as the incubation period increases from 3 days to 5 days and 7 days. Among the samples, the UMR-106 osteoblast cell grown on the THA material exhibited a significantly enhanced ALP activity compared to that of the sHA, and interestingly, there was no significant difference on glass (control) on the 7th day. The higher osteoblast activity on THA may be due to confluency being reached faster for these loaded cells and to a transition from a proliferation phase to a differentiation state phase. In addition, as reported by other researchers on the ALP activity, the results can also be attributed to the changes occurring during the amorphous/crystalline-phase transitions that occur in these substrates. The increased formation of the nanosized particles would increase the surface areas within the substrate. This would eventually lead to an improvement in the cell–biomaterials interactions [2].

The effects of the synthesized particles on zebrafish embryo survival and hatching were measured to assess the safety of sHA and THA particles on zebrafish embryo development. 24 hpf zebrafish embryos were exposed to each of these nanoparticles at the concentration of 0, 20, 40, 60, 80, and 100 µg/mL for up to 72 hpf. The cumulative survival rates of zebrafish embryos at 72 hpf for sHA and THA particles are shown in Figure 10a and Figure 10b, respectively. The survival rates of zebrafish embryos after being treated with both particles did not show a significant difference compared to the control group. Hatching is an important process during embryogenesis. The hatching rate is used for evaluating the embryotoxicity in zebrafish embryos [28]. In this study, zebrafish in the control group exposed to sHA and THA started to hatch at 48 hpf and reached a 100% hatching rate at 72 hpf (Figure 10c,d). Compared to the control, the hatching rates were found to be increased in both sHA and THA at 48 hpf. However, at 48 hpf, there was an obvious significant hatching delay for zebrafish embryos when exposed to sHA (<20%) as compared to those of THA (<60%) at all concentrations (20–100 µg/mL). This is seen in Figure 10c and Figure 10d, respectively.

To evaluate the effects of the particles on behavior and morphology of zebrafish, microscopic images of the zebrafish embryos/larvae were taken. Representative images of the embryos/larvae at 24, 48, and 72 hpf after exposure to sHA and THA particles were taken and are shown in Figure 11 and Figure 12, respectively. No malformations of zebrafish embryos/larvae were observed in the sHA and THA-treated groups. The heart rates of the zebrafish embryos were detected at 24 and 48 hpf (Figure 13). At 24 hpf, both the sHA and THA particles caused significant increases in the heart rates at higher concentrations (60, 80, and 100 µg/mL) when compared to those of the control group (shown in Figure 13a). At 48 hpf, the sHA nanoparticles significantly increased the heart rates for some concentrations (20, 60, 100 µg/mL) compared to the control group (presented in Figure 13b). On the other hand, the THA increased the heart rate up to 24 hpf, but embryos showed normal heart rates at 48 hpf (Figure 13b). These results indicated that sHA nanoparticles created more abnormal metabolic activity than THA particles. The conclusion on the cytotoxic effect of the as-prepared THA material from the in vivo zebrafish assay done in this study is that after the exposure to both sHA and THA particles, the trace element doping did not induce mortality or malformation of zebrafish embryos/larvae. In a previous study, Zhao et al. [15] investigated the biological effects of hydroxyapatite nanoparticles on zebrafish embryos and found them to cause malformations and hatching delays in zebrafish embryos/larvae. Pujari-Palmer et al. [29] also reported that hydroxyapatite particles induced mortality and malformations of zebrafish embryos/larvae. Both studies showed that the hydroxyapatite particles affected the zebrafish embryo development which is different from the results of the present study. The reasons might be due to the difference in the size or shape of the as-prepared THA (micro size with a rod or needle-like morphology) which has an effect on the degree of internalization of material constituents inside embryos [15,29].

## 4. Conclusions

In this study, the preparation of diluted ions-doped HA (THA) was synthesized via the co-precipitation method. The physical properties of the as-prepared THA particles showed the dominant characteristic properties which are different from those of the sHA (control), i.e., the crystal structure, the morphology, the surface area, the surface charge as well as the intrinsic magnetic property. These properties are strongly affected by the ions doping during the synthesis process. In addition, THA showed a faster ion release than sHA.

The biological studies revealed that all substrates did not show any cytotoxic effect on rat osteoblast-like UMR-106 cells. The UMR-106 cells had good adhesion and were better dispersed on the THA surface than those grown on the undoped HA (sHA) at each time point. A similar result was seen in the ALP activity, i.e., a significantly increased osteoblastic activity of the osteoblast-like UMR-106 cells grown on the THA substrate compared to those grown on the sHA (*p* < 0.05); an almost 1.5-fold increase was observed.

The in vivo safety assessment based on the use of zebrafish showed that the zebrafish embryo survival, hatching, and zebrafish morphology were not significantly different after exposure to sHA and THA particles. In the study on the effects of the particles on zebrafish morphology, no malformations of zebrafish embryos/larvae were observed at any time.

From the results obtained in this study, the loading of small amounts of the multi-trace metals (Mg, K, B, Fe, Zn, Mn, Cu, Ni, Mo, Sr, BO_3_^3−^, and CO_3_^2−^) into hydroxyapatite (HA) clearly stimulated the growth of the bone cells on the surface of sHA, and based on the in vivo tests conducted, the THA particles were safe without any toxic effects on embryonic zebrafish. Therefore, the physical and biological results of this study indicate that THA is a potential biomaterial candidate that could be used for bone tissue applications.

## Figures and Tables

**Figure 1 nanomaterials-13-00255-f001:**
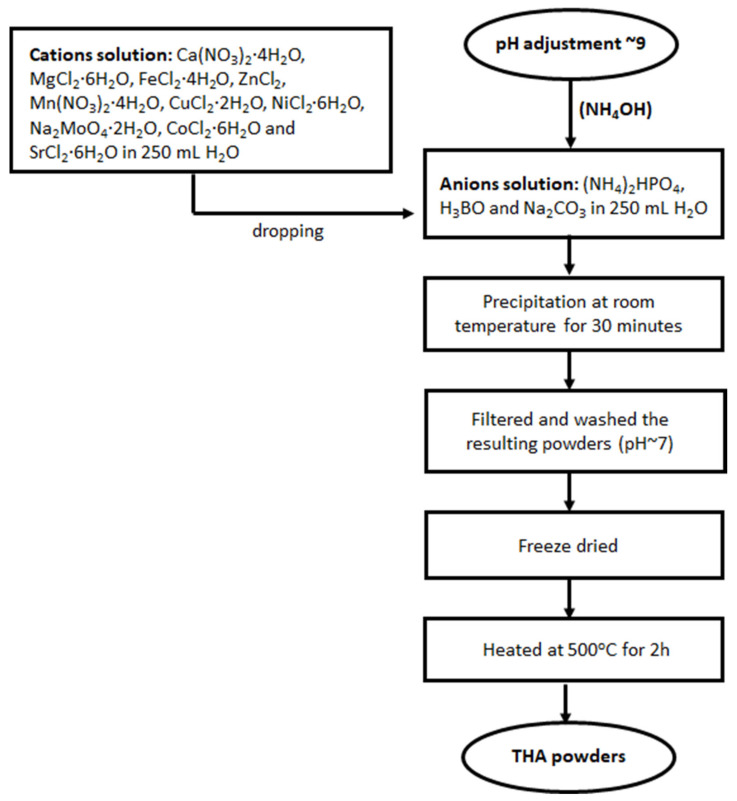
Scheme for multi-trace elements co-doped hydroxyapatite (THA) powders obtained by precipitation reactions.

**Figure 2 nanomaterials-13-00255-f002:**
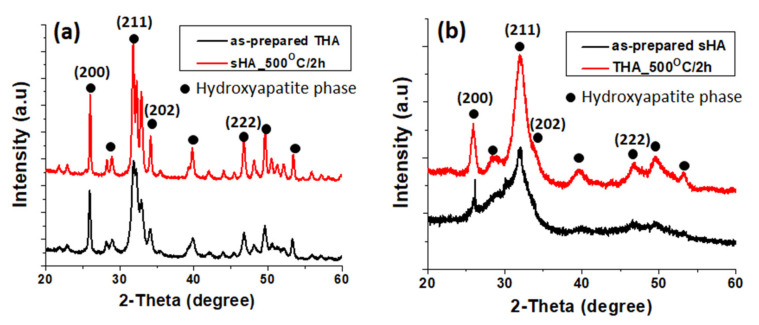
XRD patterns of sHA (**a**) and THA (**b**) powders.

**Figure 3 nanomaterials-13-00255-f003:**
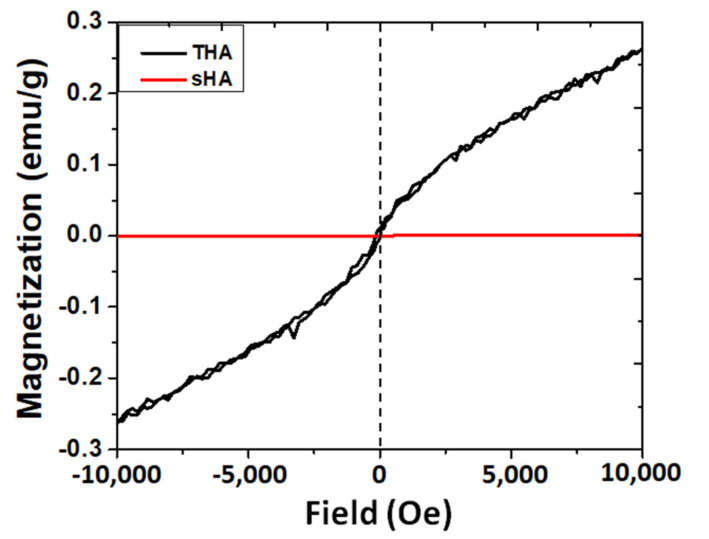
Magnetization dependence on magnetic field strength.

**Figure 4 nanomaterials-13-00255-f004:**
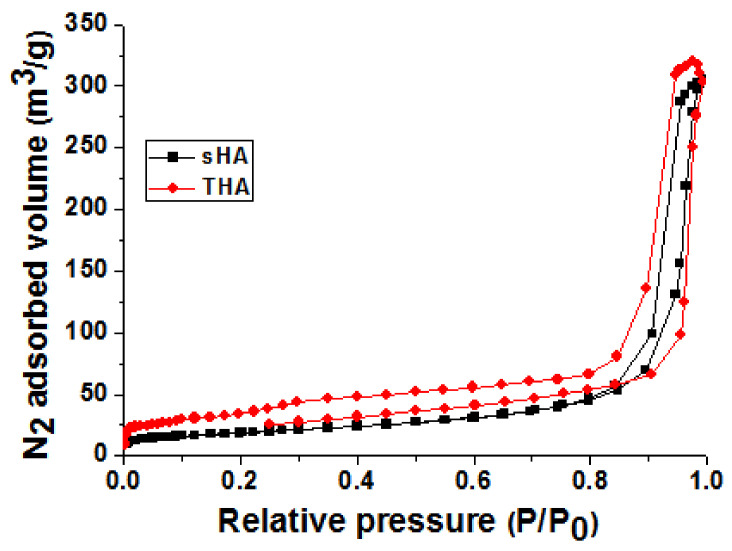
Nitrogen adsorption–desorption isotherms of sHA and THA.

**Figure 5 nanomaterials-13-00255-f005:**
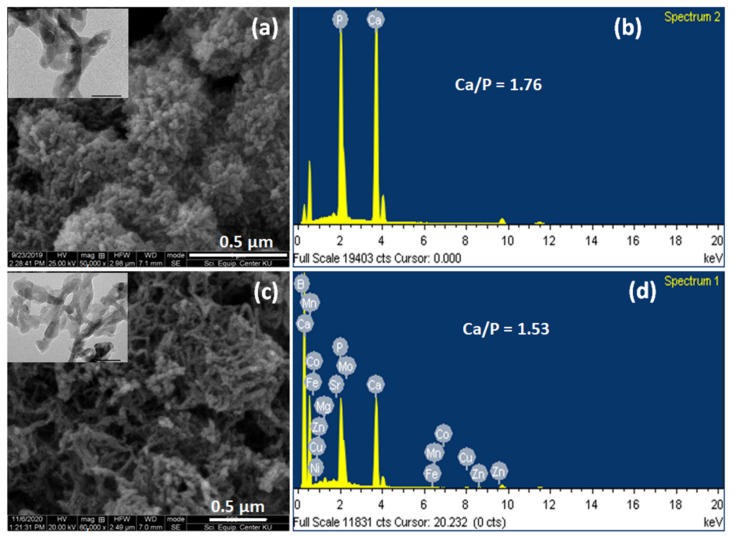
SEM (TEM inset) images (**a**) and EDS analysis (**b**) of sHA particles; (**c**,**d**) represent SEM (TEM inset) and EDS analysis of THA particles, respectively.

**Figure 6 nanomaterials-13-00255-f006:**
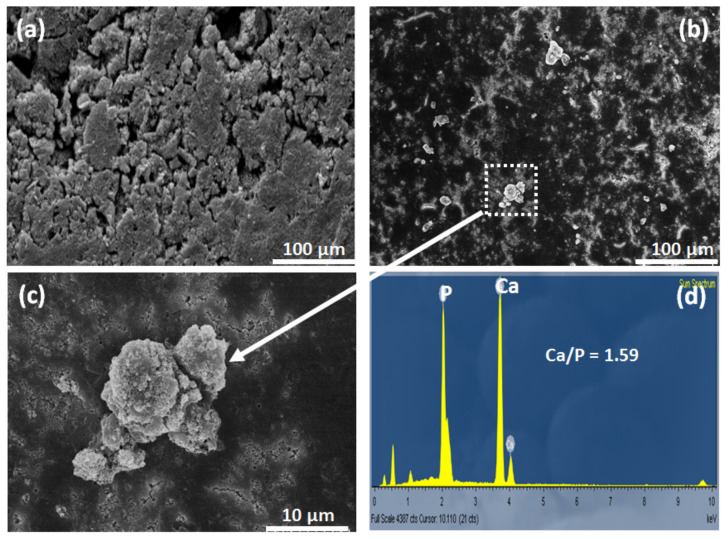
SEM image of THA surface before (**a**) and after (**b**) immersing the scaffolds in SBF for 7 days; (**c**) reveals the magnified view of the precipitation showing the accumulative structure of the mineral apatite particles and EDS analysis (**d**).

**Figure 7 nanomaterials-13-00255-f007:**
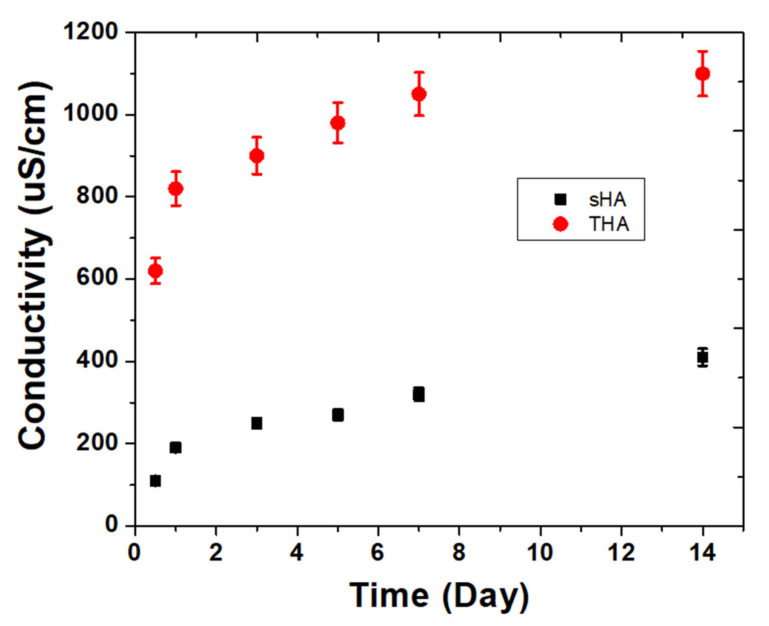
Electrical conductivity of the ions released from sHA and THA particles.

**Figure 8 nanomaterials-13-00255-f008:**
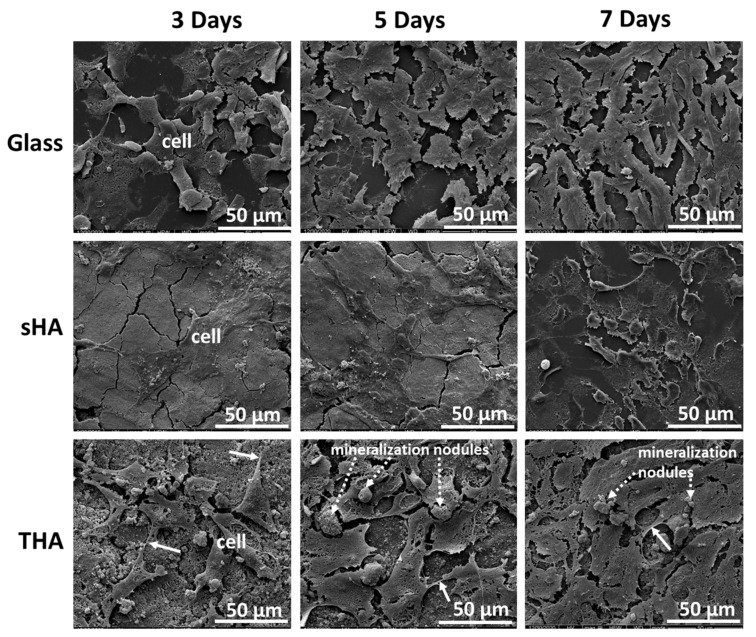
SEM images of osteoblast cells on glass, sHA, and THA surfaces after 3, 5, and 7 days of culture.

**Figure 9 nanomaterials-13-00255-f009:**
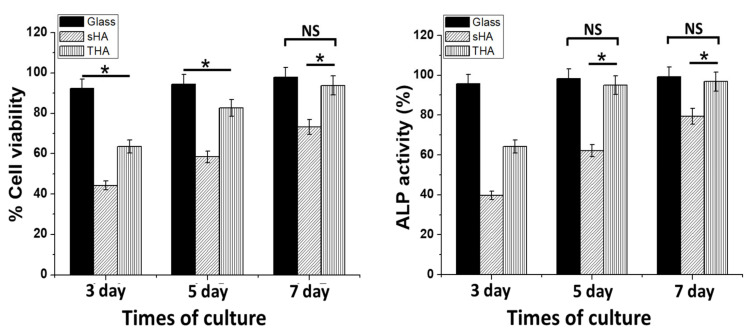
MTT assay and the relative alkaline phosphatase (ALP) activities of sHA and THA compared with the glass substrate (positive control; paired with each material) on days 3, 5, and 7. The control values were normalized to 100%. * *p* < 0.05 compared with the corresponding control group (mean ± SE). NS: no significance between the two groups.

**Figure 10 nanomaterials-13-00255-f010:**
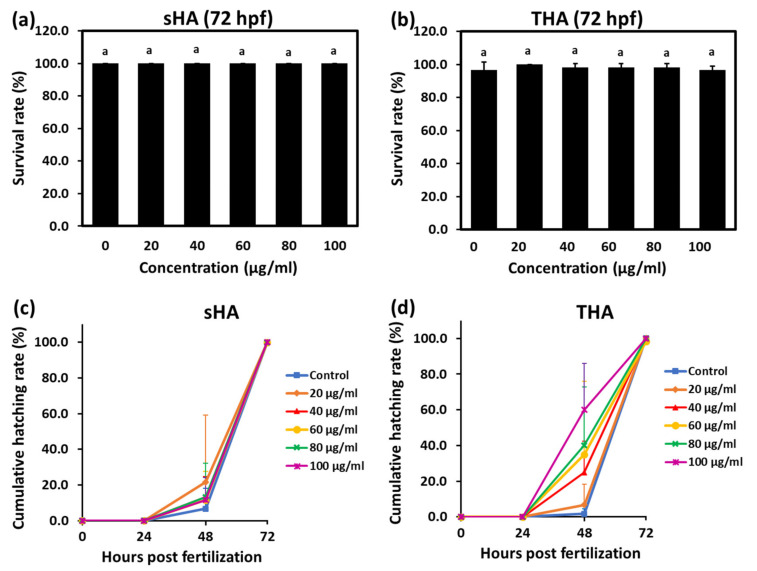
Survival rates for zebrafish embryos exposed to different concentrations (μg/mL) of sHA (**a**) and THA (**b**) at 72 h post-fertilization (hpf). Cumulative hatching rate of zebrafish embryos exposed to different concentrations of sHA (**c**) and THA (**d**) at 24, 48, and 72 hpf. The results are presented as mean ± SE (n = 3). a: *p* < 0.01 versus control.

**Figure 11 nanomaterials-13-00255-f011:**
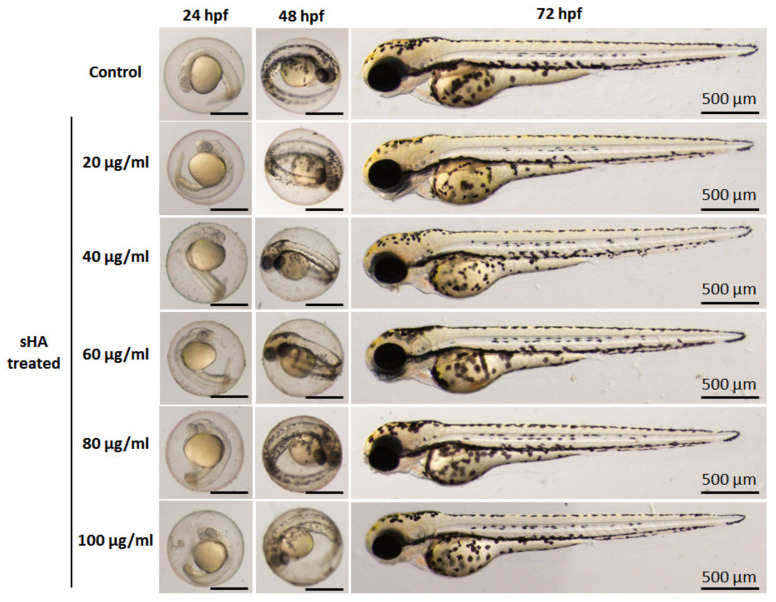
Digital photographs of zebrafish embryos at different stages of growth in solutions exposed to different concentrations (μg/mL) of sHA powders at 24, 48, and 72 hpf.

**Figure 12 nanomaterials-13-00255-f012:**
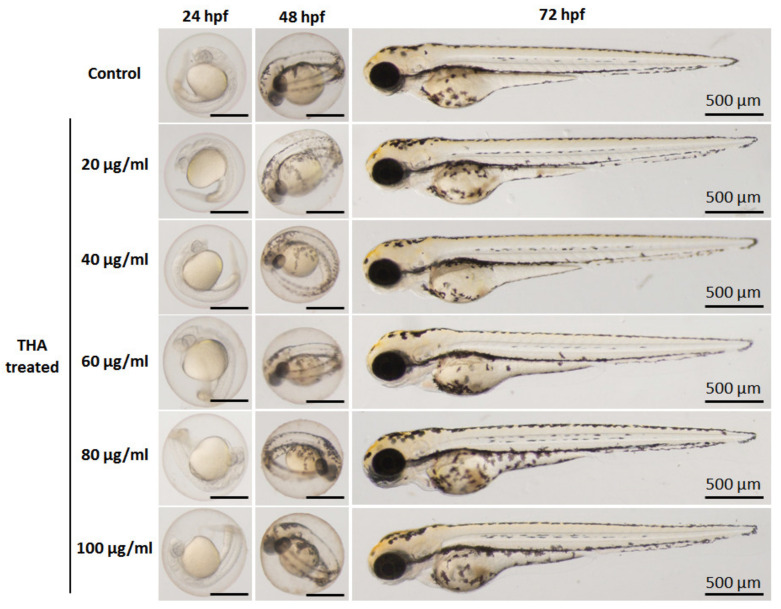
Digital photographs of zebrafish embryos at different stages of growth in solutions exposed to different concentrations (μg/mL) of THA powder at 24, 48, and 72 hpf.

**Figure 13 nanomaterials-13-00255-f013:**
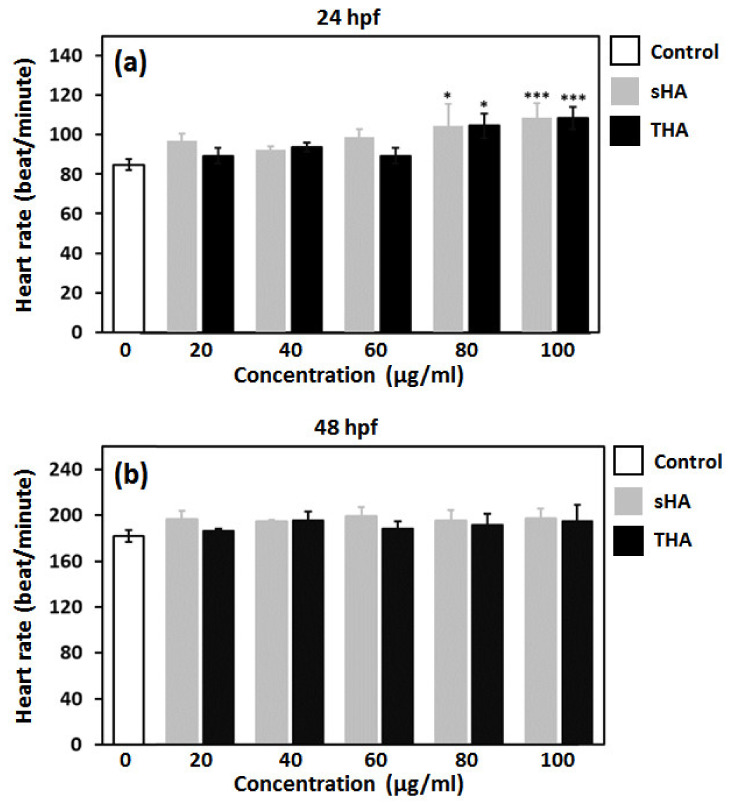
Effect of sHA and THA powders at varying concentrations on the heart rates of zebrafish embryos at 24 hpf (**a**) and 48 hpf (**b**). The results are presented as mean ± SE (n = 3), * *p* < 0.05 *** *p* < 0.001 versus control.

**Table 1 nanomaterials-13-00255-t001:** The molar concentrations of the elements used in the preparation of pure (sHA) and doped hydroxyapatite (THA).

	Cations										Anions		
Sample	Ca	Mg	Fe	Zn	Cu	Mn	Ni	Mo	Sr	Co	P	B	CO_3_
	(M)	(mM)	(mM)	(mM)	(mM)	(mM)	(mM)	(mM)	(mM)	(mM)	(M)	(mM)	(mM)
sHA	0.40	0	0	0	0	0	0	0	0	0	0.24	0	0
THA	0.38	0.50	0.075	0.08	0.02	0.006	0.001	0.001	0.008	0.001	0.23	0.01	8

## Data Availability

The datasets generated during the current work are available from the corresponding author upon reasonable request.

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
