# Peer review of "Development of Biomaterials Based on Biomimetic Trace Elements Co-Doped Hydroxyapatite: Physical, In Vitro Osteoblast-like Cell Growth and In Vivo Cytotoxicity in Zebrafish Studies"

_nanomaterials, 2023, doi:10.3390/nano13020255_

Round 1

Reviewer 1 Report

Overall, the manuscript is well designed and written well, but the authors carried out in vivo study in Zebra Fish, the ethical clearance is not mentioned in the manuscript. I recommend the manuscript may be considered after MINOR REVISION.

The authors evaluated the preparation of co-doped Hydroxyapatite sHA and THA, they have characterised the samples by various techniques and bioactivity, cell culture and in vivo study in Zebrafish embryo acute toxicity test.

It covers the gap in the relevant field.

Zebrafish embryo acute toxicity test is the novelty in this paper, since most of the researchers published about in vivo study in RAT model.

Methodology is good, no improvement is required.

The conclusion is well written.

References is restricted to only 26, the authors can include some more references.

Only one table is included with respect to Molar concentration of the elements used in the preparation of pure HAP, but some figures they can make as colour image to make it more interesting.

Author Response

Response to Reviewer 1

Overall, the manuscript is well designed and written well, but the authors carried out in vivo study in Zebra Fish, the ethical clearance is not mentioned in the manuscript. I recommend the manuscript may be considered after MINOR REVISION. 

The authors evaluated the preparation of co-doped Hydroxyapatite sHA and THA, they have characterised the samples by various techniques and bioactivity, cell culture and in vivo study in Zebrafish embryo acute toxicity test. It covers the gap in the relevant field.

Zebrafish embryo acute toxicity test is the novelty in this paper, since most of the researchers published about in vivo study in RAT model.

-Methodology is good, no improvement is required.
Response to Reviewer comment: Thank you for your comments.

-The conclusion is well written.                                                                                                                    Response to Reviewer comment: Thank you for your comments.

-References is restricted to only 26, the authors can include some more references.
Response to Reviewer comment: Thank you for your comments. Some more references related to ions-doped HA and Ca-deficient hydroxyapatite synthesis have been added in the manuscript as shown in introduction on Page 2, Line 86 and 88 and new 3 references (16, 17 and 18) have been added in the manuscript.

-Only one table is included with respect to Molar concentration of the elements used in the preparation of pure HAP, but some figures they can make as colour image to make it more interesting.                                                                          Response to Reviewer comment: Thank you for your comments. Besides table respect to molar concentration, the flowchart of the preparation process has been added in the manuscript as seen in Figure 1 on Page 4.

Reviewer 2 Report

Dear Authors,

The article is generally good and there are actually minor shortcomings that need to be filled. The idea of those zebra fish is cool. Below is a bulleted characterization of your article.

Strengths

- cool topical subject matter,

- good research methods,

- very good graphic design,

- interesting conclusions.

Weaknesses

- isolated deficiencies in the text that hinder its reception: no CAS number for reagents, no process diagram (even the simplest one), state why you chose for HAp and solutions the molar concentration?

- why do you use an angle up to a value of 60 degrees for XRD?

- as far as the SEM is concerned - the photos could be larger because you can't see much on them - and it's worth adding the type of detector and current everywhere,

- you have good conclusions, but it reads very badly - try to bullet the best conclusions, so that the reader has clear data,

- as far as literature data is concerned you are relying on quite old articles - look for something from 2019 onwards - and maybe write something about HAp in new applications in 3D printing, among other things, for jumpsuits; then you have sample articles, read them , draw conclusions, cite and add something to yourself as well:

(1) Ca-deficient hydroxyapatite synthesis on the bioapatite bovine bone substrate study

(2) Alkali-Treated Alumina and Zirconia Powders Decorated with Hydroxyapatite for Prospective Biomedical Applications

Alkali-Treated Alumina and Zirconia Powders Decorated with Hydroxyapatite for Prospective Biomedical Applications.

Best Wishes

Reviewer

Author Response

Response to Reviewer 2

The article is generally good and there are actually minor shortcomings that need to be filled. The idea of those zebra fish is cool. Below is a bulleted characterization of your article.

Strengths

- cool topical subject matter,                                                                                    - good research methods,                                                                                        - very good graphic design,                                                                                      - interesting conclusions.                                                                                         Response to Reviewer comment: Thank you for your comments.

Weaknesses

- isolated deficiencies in the text that hinder its reception: no CAS number for reagents, no process diagram (even the simplest one), state why you chose for HAp and solutions the molar concentration?                                                  Response to Reviewer comment: Thank you for your comments. The %weight of reagents and the process diagram for preparing sHA and THA substrates have been added in the manuscript as seen in Figure 1 on Page 4.

- why do you use an angle up to a value of 60 degrees for XRD?                            Response to Reviewer comment: Thank you for your comments. The XRD pattern was recorded in the range 2q = 20-60⁰ due to the remarkable sharpening peaks of hydroxyapatite can be observed. Furthermore, the authors wanted to explore whether they were other phases in the sample other the desired phases.

- as far as the SEM is concerned - the photos could be larger because you can't see much on them - and it's worth adding the type of detector and current everywhere,                                                                                                       Response to Reviewer comment: Thank you for your comments. The photos in the manuscript have been checked and the details operating for SEM have been added in the manuscript on Page 5, Line 165-167.

- you have good conclusions, but it reads very badly - try to bullet the best conclusions, so that the reader has clear data,                                                        Response to Reviewer comment: Thank you for your comments. The conclusion has been rewritten as seen on Page 16.

- as far as literature data is concerned you are relying on quite old articles - look for something from 2019 onwards - and maybe write something about HAp in new applications in 3D printing, among other things, for jumpsuits; then you have sample articles, read them, draw conclusions, cite and add something to yourself as well:                                                                                                      (1) Ca-deficient hydroxyapatite synthesis on the bioapatite bovine bone substrate study                                                                                                          (2) Alkali-Treated Alumina and Zirconia Powders Decorated with Hydroxyapatite for Prospective Biomedical Applications                                                                  Alkali-Treated Alumina and Zirconia Powders Decorated with Hydroxyapatite for Prospective Biomedical Applications.                                                                      Response to Reviewer comment: Thank you for your comments. The literature data about Hap in new applications and the references suggested by reviewer have been cited in this manuscript as seen in Ref. 16-18.

Reviewer 3 Report

For the most part, this manuscript provides sufficient experimental evidence to support the proposed goal and conclusions. Only minor corrections are required as described below:

How did you prepare the THA (and sHA) discs for osteoblast cell cultures? Based on Figure 7, there appears to be more THA on the surface of cell culture discs than sHA and perhaps this could have biased the attachment/proliferation of cells in the favor of THA? Please clarify.

Please use professional language when citing other authors. Please refrain from using first-person pronouns like he/she or his/her (line 78).

Minor revision of English language is recommended

Author Response

Response to Reviewer 3

For the most part, this manuscript provides sufficient experimental evidence to support the proposed goal and conclusions. Only minor corrections are required as described below:

-How did you prepare the THA (and sHA) discs for osteoblast cell cultures? Based on Figure 7, there appears to be more THA on the surface of cell culture discs than sHA and perhaps this could have biased the attachment/proliferation of cells in the favor of THA? Please clarify.                                                 Response to Reviewer comment: Thank you for your comments. In this research the authors control the size of substrates of circular glass discs (control), sHA and THA discs with a dimeter of ~10 mm and a thickness of 3 mm for cells culture test. The description has been added in the manuscript on Page 5, Line 203-204.

-Please use professional language when citing other authors. Please refrain from using first-person pronouns like he/she or his/her (line 78).                                    Response to Reviewer comment: Thank you for your comments. The corrected word has been checked on Page 2, Line 78.

-Minor revision of English language is recommended                                          Response to Reviewer comment: Thank you for your comments. The English language has been checked as seen in red letters.
